# Preterm Birth, Developmental Smoke/Nicotine Exposure, and Life-Long Pulmonary Sequelae

**DOI:** 10.3390/children10040608

**Published:** 2023-03-23

**Authors:** Chie Kurihara, Katherine M. Kuniyoshi, Virender K. Rehan

**Affiliations:** Department of Pediatrics, The Lundquist Institute for Biomedical Innovation at Harbor-UCLA Medical Center, David Geffen School of Medicine, University of California Los Angeles, Los Angeles, CA 90095, USA; ckurihara@dhs.lacounty.gov (C.K.); kkuniyoshi@dhs.lacounty.gov (K.M.K.)

**Keywords:** prematurity, preterm birth, nicotine, exposome, transgenerational

## Abstract

This review delineates the main pulmonary issues related to preterm birth, perinatal tobacco/nicotine exposure, and its effects on offspring, focusing on respiratory health and its possible transmission to subsequent generations. We review the extent of the problem of preterm birth, prematurity-related pulmonary effects, and the associated increased risk of asthma later in life. We then review the impact of developmental tobacco/nicotine exposure on offspring asthma and the significance of transgenerational pulmonary effects following perinatal tobacco/nicotine exposure, possibly via its effects on germline epigenetics.

## 1. Introduction

Preterm birth (PTB)/prematurity is “the leading cause of death in children under five years”, with approximately one million annual deaths globally. It is one of the top healthcare priorities, both in the United States and globally [1]. The United Nations Sustainable Development Goal Target 3.2 aims to end preventable deaths of newborns and children under 5 by 2030. To accomplish this, all countries are aiming to reduce neonatal mortality to at least 12 per 1000 live births and under-5 mortality to at least 25 per 1000 live births [2]. Specifically, The Healthy People initiative, spearheaded by the U.S. Department of Health and Human Service’s Office of Disease Prevention and Health [3], in its fifth iteration, Healthy People 2030 (MICH-7), aims to reduce the incidence of PTB to 9.4% by 2030 [4]. Since maternal exposure to tobacco smoke during pregnancy is a major cause of PTB [5], the second goal of Healthy People 2030 is “to increase abstinence from cigarette smoking to 95.7% by 2030.” Data from 2019 show that the current abstinence rate is at 94% [6]. Significantly, maternal smoking in pregnancy is independently associated with both preterm delivery and increased respiratory morbidities such as asthma [7,8,9]. Given smoking’s harmful effects and its association with PTB and asthma, numerous organizations are attempting to mitigate this public health crisis. However, despite all efforts, we have still not seen sustainable declines in either PTB or asthma.

## 2. Preterm Birth Rates and Recent Trends

Due to challenges in many countries with data collection, especially those that are low-income, determining the actual prevalence of PTB is difficult. The World Health Organization (WHO) estimates that there are approximately 15 million preterm (<37 weeks’ gestation) deliveries per year globally [10,11]. This results in a PTB rate of 11%, and for many countries, it continues to be on the rise [11]. Global PTB rates increased from 9.8% in 2000 to 10.6% in 2014 [12]. The United States is ranked sixth “among the top 10 countries with the greatest numbers of PTBs,” with 517,000 cases annually [13]. The top six countries with the highest prevalence—India, China, Nigeria, Pakistan, Indonesia, and the U.S.—comprise approximately 50% (~7.4 million) of total PTBs worldwide [13]. For the U.S. specifically, there have been fluctuating trends in the incidence of PTB since 2007. National Vital Statistics showed that from 2007 to 2014, the PTB rate decreased from 10.44% to 9.57%, but then increased by 7% from 2014 to 2019 [14]. There was a marginal improvement from 2019 to 2020, which was not observed in all race groups. It declined by 2% in non-Hispanic White mothers, from 9.26% to 9.10%, and by 1% in Hispanic mothers, from 9.97% to 9.84%. No significant change was noted in non-Hispanic Black mothers [15]. The most recent PTB rate in 2020 spanned from a minimum of 8.51% in non-Hispanic Asian mothers to a maximum of 14.36% in non-Hispanic Black mothers [15]. This highlights the disparities in PTB rates and difficulties with health equity. Moreover, it does not appear that the recent trend of a decrease in PTB rate will persist. For example, Martin JA recently (2021) reported an uptrend in PTB in all racial groups [16]. 

Due to improved perinatal care, including antenatal steroids, surfactant therapy, and other technological advances, there are more adult survivors of PTB now than at any time. In addition to mortality risk, prematurity is a significant risk factor for lifelong adverse sequelae, including significant pulmonary complications such as chronic lung disease and asthma [17,18,19,20,21,22,23,24,25], constituting a significant economic burden. The U.S. alone spends approximately USD 26 billion yearly on the medical care of preterm infants [26]. This does not include the cost of potential adverse sequelae, such as chronic lung diseases, including asthma and neurodevelopmental delays [10,11,12,13,27,28,29].

## 3. Normal Lung Development

Implications of PTB and tobacco/nicotine exposure on the developing lung first require a brief overview of normal lung development. Lung development involves five stages encompassing both prenatal and postnatal periods (Figure 1). The conducting system begins to develop in the fourth week of gestation, followed by the formation of the gas exchange components, which extends well into late childhood and even to early adulthood [30,31,32]. During the *embryonic stage* (gestational weeks 4–7), two lung buds develop from the ventral wall of the primitive foregut. These elongate and initiate branching (branching morphogenesis) [33], which is finely orchestrated by epithelial-mesenchymal crosstalk [34,35]. The initial pulmonary vessels (vasculogenesis) are formed as a plexus in the mesenchyme surrounding the lung buds, with the formation of a new capillary plexus (angiogenesis) surrounding each newly formed bud with further branching [36,37]. The second *pseudoglandular stage* (weeks 7–17) primarily involves the formation of the first 20 generations of airway branching [38], the first few generations of alveolar ducts, and the appearance of proximal airway cells [39]. The third *canalicular stage* (weeks 17–27) includes the differentiation of the epithelium, allowing the morphological distinction between conducting and respiratory airways [40]. By the end of this stage, the alveolar epithelium comes in close proximity with the mesenchymal capillary network, allowing gas exchange in babies born extremely premature [37]. During the fourth *saccular stage* (weeks 24–38), the terminal airways grow in length, widen, and form clusters of larger airspaces, i.e., saccules, and the gas-exchanging (type 1) and surfactant-producing (type 2) epithelial cells differentiate further [37]. The fifth and final *alveolar stage* (weeks 36 and beyond) is the longest lung development stage that can continue to approximately 21 years of age, although most alveolarization is achieved by 8 years [41,42]. 

Although infants born extremely preterm, i.e., during the late canalicular and early saccular stages of lung development, have the most significant burden of early respiratory morbidity that predisposes them to the most significant risk for later pulmonary issues, a complex, intricate and protracted lung development process, starting from the early embryonic period and continuing into early adult years, renders the respiratory system to be vulnerable to insults throughout its developmental course. Furthermore, although the lung’s functional capacity increases throughout childhood and peaks at 20–25 years before starting its decline into old age, it is well established that the trajectory of its maturation is already established at birth. For example, maximal expiratory flow measured shortly after birth correlates with the forced expiratory volume in one second/forced vital capacity (FEV1/FEC) ratio into adult life [43]. In addition to the interruption of natural lung development due to PTB, the damaging effects from inadequate nutrition, respiratory infections and respiratory support, such as oxygen supplementation and invasive or non-invasive ventilatory support that extremely premature infants universally require for their extrauterine survival, adds to both short- and long-term respiratory consequences (Figure 1) [44]. These invariably include sub-optimal adult respiratory capacity, an increased predisposition to asthma, and accelerated decline in lung function, although the data are inconsistent between various studies. Furthermore, as outlined later, exposure to tobacco smoke/nicotine at any stage in the developing lung is detrimental [45].

## 4. Preterm Birth and Life Long Respiratory Sequelae

With an increase in PTB rate, there has also been an increase in survivors of PTB of up to 95% [46,47] into adult years. This has been accompanied by increased chronic health problems among the survivors. These include, but are not limited to, chronic cardiovascular, metabolic, endocrine, renal, central nervous system, and pulmonary disorders such as asthma and other chronic lung diseases [7,18,23,44,48,49,50,51,52]. Although heterogeneity among study populations, advances in perinatal and neonatal care over time, and numerous other confounding variables (e.g., genetic background, socioeconomic status, and tobacco exposure) makes a direct comparison between various studies challenging, in general, as reviewed next, the pulmonary outcomes at all ages following PTB are compromised compared to those born at term [44,53]. 

## 5. Preterm Birth and Respiratory Symptoms at Various Life Stages

During early childhood and preschool years, children born preterm, both with and without bronchopulmonary dysplasia (BPD), show more respiratory symptoms, such as cough and wheezing, than those born at term [54,55,56]. Similarly, during school-age years, former preterm infants compared to term-born controls demonstrate more significant respiratory symptoms (wheezing and cough) independent of their BPD status [53,57,58,59,60], although there is a higher frequency of symptoms in those with BPD versus those without BPD [61]. In contrast to the unequivocal evidence of adverse respiratory outcomes in preschool and school-age former preterm-born children, the data in the adolescent age group are somewhat limited and inconclusive. Both higher [62] and equal [63] prevalence of respiratory symptoms have been reported in former preterm-born (versus term-born control) adolescents. Like the preschool and school-age groups, prematurely born young adults, compared to those born at term, show more respiratory symptoms irrespective of their BPD status [64,65,66]. Although some studies suggest that symptoms, such as cough, wheezing, long-term use of respiratory medications, and impaired quality of life, remain more frequent in former preterm-born individuals with BPD versus those without BPD, in general, there is an overall decrease in the frequency of respiratory symptoms from early childhood to adulthood in both with and without BPD populations [67].

## 6. Impact of Preterm Birth on Exercise Tolerance

The data on exercise capacity in former preterm-born individuals is in line with the data on respiratory symptoms, i.e., lower exercise capacity in those with BPD (versus non-BPD and term controls) [66,68,69,70] and without BPD (vs. term controls) [44,68,69]. In a select cohort of former preterm-born individuals from a large population-based Swedish database, PTB was noted to be an independent predictor of reduced exercise capacity [71]. The reasons underlying reduced respiratory function in former preterm-born adults are likely to be multifactorial, e.g., lower lung volumes, neonatal respiratory support-related lung damage, more frequent respiratory infections, and respiratory de-conditioning related to reduced activity, contributing to variable degrees in different populations. Lung volumes in premature infants born before 28 weeks gestation with or without BPD when measured at term equivalent are lower than those born at term [72]. This likely contributes to continued lower respiratory capacity over time. For example, in a study involving preterm children, as assessed by spirometry, worsening lung function was observed at ages 3–6, 7–11, and 12–20 years [73]. In general, prematurely delivered individuals with or without BPD do not achieve optimal peak lung function as adults and have decreased lung function despite receiving antenatal corticosteroids and surfactant BPD [19,73,74].

## 7. Preterm Birth and Predisposition to Airway Hyperresponsiveness/Asthma

A high prevalence of airway hyperresponsiveness/asthma following PTB has been found at all postnatal ages examined [49,75,76,77,78,79]. The overall wheezing risk in former preterm-born individuals is estimated to be increased by 1.71-fold with an inverse correlation with gestational age at birth [80]. However, due to heterogeneity in the study populations, the definitions of wheezing and asthma used, and variations in exposure to other factors, such as pollution, maternal smoking, and hygiene conditions, which are all known independent contributors to wheezing disorders, the actual risk of hyperresponsiveness/asthma varies. In general, higher rates of wheezing in premature children, both with and without BPD, likely account for high asthma rates in this population. However, whether wheezing is largely reversible or not, it helps differentiate asthma from prematurity-related lung parenchymal structural damage and or airway wall thickening that lead to fixed airflow obstruction and wheeziness [81]. Lower concentrations of exhaled nitric oxide and exhaled breath temperature and the absence of eosinophilic airway inflammation in wheezy former preterm-born children with or without BPD versus children with asthma suggest different underlying mechanisms resulting in wheezing under these two scenarios [82,83,84]. However, evidence of ongoing oxidative stress in former preterm-born individual’s airways suggests an ongoing airway disease and not just stabilized structural lung damage from preterm birth and other accompanying insults [85]. Additionally, subglottic stenosis and tracheomalacia/bronchomalacia secondary to prolonged mechanical ventilation and left recurrent laryngeal nerve palsy after surgical ligation of patent ductus arteriosus might also result in fixed airway obstruction and wheezing in former preterm-born individuals that are unresponsive to bronchodilators [86]. Intrauterine growth restriction, often accompanying preterm birth, also enhances airway obstruction, manifesting clinically as persistent wheezing [87].

## 8. Preterm Birth and Predisposition to Respiratory Infections

Due to pulmonary and immune immaturity, PTB is associated with increased respiratory infections in infancy and early childhood; however, whether this predisposition continues into adolescence and adulthood is unclear [88,89,90]. In a national cohort of all infants born in Denmark from 1992 to 2007, although higher rates of respiratory infections and higher odds of hospitalization for airway infections were found in former preterm-born children (GA 23–27 weeks) at 4–5 years of age, there was no difference in these rates in adolescents [88]. Low birth weight has also increased the risk of severe SARS-CoV-2 infection in non-elderly adults [91]; however, whether PTB has an independent association remains to be proven [92]. 

Once a preterm delivery has occurred, it can be challenging to mitigate the cascade of events that ultimately result in morbidities, the degree to which is dependent upon the gestation: the lower the gestation at birth, the higher the risk of morbidities and mortality. Moreover, higher morbidities in childhood set the stage for higher adulthood adverse events, as described by Barker [93]. Therefore, the prevention of PTB may be the key to the prevention of many of these adulthood conditions. There is a myriad of causes for prematurity: inflammation and infection, gestational diabetes, maternal hypertension, multiple gestations, advanced maternal age, maternal obesity, in vitro fertilization, and smoke exposure [94,95,96,97,98,99,100,101,102,103,104,105,106]. Although many of these risk factors are unmodifiable, others, such as decreasing or abstaining from tobacco/nicotine usage during pregnancy, can make a significant difference. Importantly, maternal smoking in pregnancy is independently associated with both preterm delivery and lifetime predisposition to respiratory illnesses, including asthma [7,8,9], rendering prevention of perinatal smoke exposure one of the top public health priorities. An estimated 20% of healthcare costs (USD ~1 billion annually) for childhood respiratory illnesses are directly attributable to maternal smoking [107].

## 9. Effects of Perinatal Tobacco/Nicotine Exposure on Developing Lungs

Based upon the 2020 National Health Interview Survey, an estimated 47.1 million U.S. adults (19.0%) reported currently using any commercial tobacco product, including cigarettes (12.5%), e-cigarettes (e-cigs, 3.7%), cigars (3.5%), smokeless tobacco (2.3%), and pipes (1.1%) [108]. There is evidence that despite warnings from the Food and Drug Administration and other agencies, there has recently been a marked increase in the use of e-cigs among women of reproductive age [109]. Regardless of the form by which these are consumed, tobacco products and e-cigs contain thousands of compounds (toxicants, carcinogens, and nicotine) [110] that play significant roles in the adverse health consequences of their users [111], as well as the fetus [112,113], and even subsequent generations [114,115,116]. However, convincing evidence supports that smoke exposure’s effects on the developing lung are predominantly caused by nicotine [114,117,118]. It crosses the placenta with minimal biotransformation and accumulates in amniotic fluid and several fetal tissues, including the respiratory tract [119,120]. Since, in comparison to maternal concentrations, fetal serum nicotine concentrations can be higher by 15% [121], the fetus is exposed to even higher nicotine than the smoking mother. Notably, the pulmonary phenotype seen following perinatal nicotine exposure in multiple experimental models is similar to that seen in human infants exposed to maternal smoke during pregnancy [118,122,123]. In a multicenter study, subjects born to mothers who smoked during pregnancy had an adjusted hazard ratio of 1.79 [95% CI, 1.20–2.67] for increased risk of asthma compared to those born to non-smoking mothers [124]. In an earlier groundbreaking study, Tager et al. [125] established that even sidestream smoke affected fetal lung development and neonatal pulmonary function. 

It is widely established that there are critical periods in an individual’s lifespan when the impact of environmental exposures is at its peak. These vulnerable windows include conception (where plasticity can be at its maximum) [126], birth, puberty, pregnancy, and menopause [127]. The perinatal period, defined as the period from 20 completed weeks of gestation to 28 postnatal days, or defined alternatively as the period starting from 22 completed weeks of gestation and lasting up to seven days after birth [128], is particularly vulnerable to the detrimental effects of tobacco smoke/nicotine. During this time frame, nicotine particularly affects cellular differentiation and conducting airways [129], explaining the reduced respiratory function in the premature infants of smoking mothers compared to premature infants of nonsmokers [130]. In vitro studies have demonstrated that nicotine directly affects alveolar type II cells, fibroblasts, endothelial cells, and stem cells, among the many other lung cell types affected [131,132,133]. These effects likely explain the dysanaptic lung growth and branching, increased collagen deposition, thicker alveolar walls, airway smooth muscle volume, altered tidal volumes, and airflow restriction observed in various rodent, sheep, and primate models of gestational and perinatal nicotine exposure [118,123,129,130,134,135]. Animal models have also demonstrated arrested lung growth and hypoplasia [131,134,136] with larger and fewer saccules that are more compliant, have fewer septal crests, reduced parenchymal tissue, and decreased gas exchange area secondary to overall hypoplastic lungs of smoke-exposed fetal rats [137]. Interestingly, males demonstrated a more significant asthmatic pulmonary phenotype in a well-established rat model of perinatal nicotine exposure [138]. 

Airway responsiveness following perinatal smoke/nicotine exposure is primarily driven by nicotine’s effect on the developing mesenchyme. During development, a delicate balance between the mesenchymal Wingless/Int (Wnt) and peroxisome proliferator-activated receptor γ (PPARγ) signaling controls airway smooth muscle cell and alveolar fibroblast differentiation [139,140]. Acting via nicotinic acetylcholine receptors (nAChRs), expressed extensively by bronchial epithelial, endothelial, mesenchymal, and pulmonary neuroendocrine cells during early fetal life [141], perinatal nicotine exposure upregulates Wnt signaling and downregulates PPARγ signaling pathways in the developing lung (Figure 1) [138,140,142], resulting in excessive myofibroblast differentiation and proliferation, a hallmark of airway hyperresponsiveness seen in perinatal smoke/nicotine-exposed children and adults. Nicotine exposure results explicitly in upregulation of α7nAChR in airway fibroblasts, followed by an increase in collagen and a decrease in elastin deposition [123,143]. These molecular and cellular phenomena likely represent the main mechanisms underlying perinatal smoke/nicotine exposure-induced lung phenotype. However, limited animal and human studies have also suggested perinatal smoke/nicotine-induced altered immune alterations such as Th2 polarization [144,145,146] and enhanced responses to postnatal allergen and fungal exposures following in utero smoke exposure [147].

## 10. Epigenomics of Smoke/Nicotine Induced Pulmonary Programming

Epigenomics involves modifying gene expression through epigenetic processes that alter the genome without affecting the underlying DNA sequence. These processes include DNA methylation, histone modifications, or RNA silencing, resulting in changes that can persist through cell division and be inherited through multiple generations [148]. Therefore, in contrast to an individual’s genome, which is constant, the epigenome is a dynamic response to external exposures and stimuli that alters how genes are expressed [149]. Although roles of both heritability and epigenetics have been studied extensively in asthma and chronic obstructive pulmonary disease pathogenesis [150,151,152], accumulating evidence suggests a more dominant role played by epigenetic factors [153,154,155,156,157,158,159,160]. 

Several studies have examined epigenetic programming following perinatal smoking/nicotine exposure. Gene and tissue-specific DNA methylation changes have been described in the placenta, fetus, and cord blood after prenatal smoke exposure [161,162,163,164], and the persistence of some of these changes on specific loci has been shown up to adolescence on buccal and blood analysis [165,166,167]. Interestingly, changes at specific loci have been linked to specific morbidities, including an increased predilection for asthma [168,169]. For example, specific genes such as AHRR, GF1, FOXP3, CYP1A1, and RUNX that may play a role in childhood respiratory morbidities were identified in cord blood studies [162,168,169,170,171,172,173]. Intriguingly, RUNX1 polymorphism is directly linked to airway hyper-responsiveness and asthma, and after in utero nicotine exposure, its expression increases in the human embryonic lung [168]. Additionally, placenta and cord samples from pregnancies exposed to tobacco were noted to have hypermethylation in RUNX1 and RUNX3 [162,172] and hypomethylation in AHRR loci in the cord blood of human neonates, correlating with maternal cotinine concentration and persisting up to 18 months of age [166,174,175]. Finally, deletions or structural polymorphisms in CYP1A1 and GSTT1 enzymes, involved in detoxification and metabolism processes of the tobacco products’ toxic metabolites, were found to be associated with increased asthma risk and sensitivity of the fetus to maternal smoking [165,176]. 

## 11. Transgenerational Inheritance via Smoking/Nicotine’s Effects on Germline Epigenetics

Many studies have shown that not only the first-generation offspring are at risk following maternal exposure to smoke/nicotine during pregnancy, but the progeny of subsequent generations might also be at risk for pulmonary complications even without any subsequent exposure to smoke/nicotine. This indicates that the adverse health impact of smoking is heritable, and persistent and has been observed both clinically and experimentally. In a Southern California telephone survey, tobacco smoke exposure during pregnancy was determined to have had potential effects on the lung development of the grandchild, regardless of the mother’s smoking status during pregnancy [177]. A similar association was seen in grandchildren developing asthma among those whose grandmothers smoked during their pregnancies [178]. Interestingly, paternal studies demonstrated that nicotine exposure by the father in utero affected the respiratory outcome of his daughter, independent of his smoking habits [179]. However, not all studies support the occurrence of transgenerational asthma following prenatal smoke exposure. For example, there was no conclusive evidence for any correlation between prenatal smoking by grandmothers and any effect on grandchildren in the Avon Longitudinal Study of Parents and Children [180]. The differences in genetically diverse populations likely studied, and the difficulties in excluding the effects of numerous other confounding variables account for the observed discrepancies in the results of these studies. Overall, epidemiological and clinical studies suggest but do not prove smoke/nicotine’s transgenerational pulmonary effects. 

Given that it would take decades to follow multiple generations prospectively and since, in real life, it is almost impossible to control for all variables determining asthma, experimental models are the only realistic way to study the phenomenon of transgenerational transmission of asthma. As outlined next, that is what we and others have done. A murine model demonstrated that maternal tobacco exposure increased airway hyperactivity, airway resistance, and decreased lung compliance in offspring, which was then passed on to the next generation without tobacco exposure [145]. Even more convincingly, in an experimental rat model, airway hyperresponsiveness and asthma-related molecular markers were increased in the first-generation offspring exposed to nicotine and subsequent second and third-generation offspring without any subsequent exposure to nicotine beyond the F0 generation (Figure 2) [181,182]. Supporting this highly significant observation, nicotine-induced transgenerational inheritance of additional phenotypic and molecular traits has also been described in other models [115,183]. Additionally, since the offspring of smokers are more likely to smoke than nonsmokers [184,185], the effect of repeat smoke/nicotine exposure during a subsequent generation following exposure in F0 gestation on the asthmatic phenotype in F2 generation has been examined [114]. It was determined that nicotine exposure in F1 gestation following its exposure in F0 gestation causes a more robust asthma phenotype in the F2 generation, especially in males, compared to its exposure in only F0 pregnancy.

While the exact mechanism(s) underlying any transgenerational inheritance remains unknown, epigenetic inheritance via alterations in germline DNA methylation, histone acetylation, and small RNAs have been implicated as plausible mechanisms [114,181,186,187]. Though most environmental exposures-induced epigenetic changes are known to be discarded at every generation, there is irrefutable evidence of germline reprogramming [188], retentions of heritable epigenetic marks, and the transmission of the environmentally induced epigenetic marks across generations in a variety of models [189,190,191,192], rendering smoke/nicotine-induced germline epigenetic inheritance as a potential plausible mechanism (Figure 2). 

Given the above-reviewed pulmonary consequences and other well-known harms of perinatal smoke exposure, many organizations and campaigns have focused on tobacco cessation [193,194,195,196]. Although there has been improvement, the WHO reported that there are still approximately 1.3 billion smokers globally, contributing to 8 million deaths yearly [197]. Specifically in the U.S., 500,000 smokers die prematurely yearly, and an additional 16 million develop significant morbidities, accounting for USD 225 billion in healthcare costs yearly [196]. Although tobacco/nicotine usage during pregnancy continues to be common, there has been a modest improvement in women smoking anytime during their pregnancy from 2016 (7.2%) to 2020 (5.5%) [15,198]. In addition, from 2005 to 2020, adults generally smoked fewer cigarettes per day. In 2020, 12.5% of U.S. adults aged ≥18 smoked cigarettes, the lowest prevalence since data became first available in 1965 [199]. These statistics are encouraging, and these improvements may be due to multi-pronged anti-tobacco interventions [193,194,195,199,200]. Although these statistics are promising, there are still challenges to its continued reduction due to multiple barriers, such as the highly addictive nature of nicotine [201], substantial withdrawal symptoms [202], and aggressive advertising strategies that target adolescents [203]. Since a dose-effect relationship exists between nicotine intake and smoking-related outcomes of pregnancy [9], smoking cessation strategies at any stage during pregnancy should be emphasized.

## 12. Conclusions

Given that adults born preterm remain at increased risk for pulmonary structural and functional compromise, respiratory infections, and respiratory health-related hospital admissions, it is imperative to ensure continued close oversight over these issues as a former preterm-born child transitions to adult care. It is also evident that the adult pulmonary outcomes of pre-surfactant and pre-antenatal steroid-a era preterm-born individuals differ from those routinely receiving these interventions. Additionally, there is no doubt that due to changing threshold of viability and the ever-improving perinatal, neonatal, and pediatric care, the outcomes of preterm-born in the future will also be different from those receiving contemporary medical care. Similarly, changing perinatal tobacco/nicotine exposure patterns in preterm-born individuals can also potentially impact the pulmonary outcomes differently. Nevertheless, despite knowing the dangers of perinatal exposure to smoking for over six decades [204], it remains a major public health concern [205]. Fortunately, various campaigns globally and nationally have prioritized mitigating tobacco/nicotine usage for the general population and pregnant women. It is gratifying to note that according to the most recent National Center for Health Statistics (NCHS) brief (#458, January 2023), between 2016 and 2021, the percentage of mothers who smoked during pregnancy declined across all maternal age groups, regardless of race or ethnicity. It occurred in all 50 U.S. states and the District of Columbia. 

Inquiry into preterm birth and perinatal health/exposures should be part of routine adult health care with an emphasis on avoidance of tobacco smoking and e-cig vaping, emphasis on regular exercise, and adopting a generally healthy lifestyle, as well as ensuring annual influenza immunization [83]. Whether routine spirometry in adult follow-up clinics will be a valuable tool in monitoring the respiratory health of prematurely born adults remains to be proven. Furthermore, bronchodilator and/or inhaled corticosteroid therapy should be based upon individualized needs and risk-benefit analysis rather than the persistent presence of audible wheeze only. With a deeper mechanistic understanding of tobacco/nicotine-mediated damage to the developing lung and its transgenerational inheritance, attention can also be placed on epigenetic-targeted therapeutics in the future for those who experience continued exposure to the peril of perinatal tobacco/nicotine. 

## Figures and Tables

**Figure 1 children-10-00608-f001:**
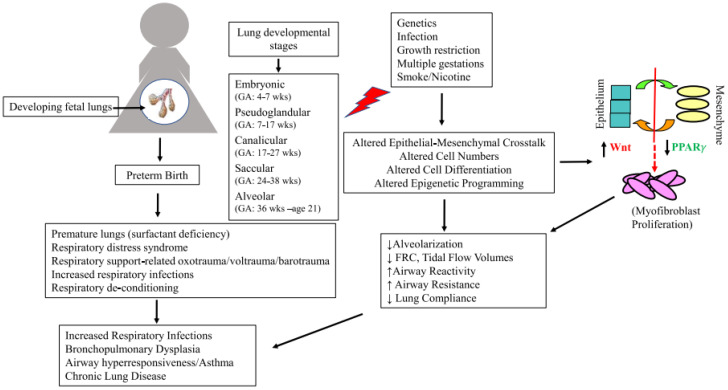
Schematic representation of normal lung development, common triggers of preterm birth, key cellular and molecular pathways involved in preterm birth- and smoke/nicotine exposure-related short- and long-term pulmonary outcomes.

**Figure 2 children-10-00608-f002:**
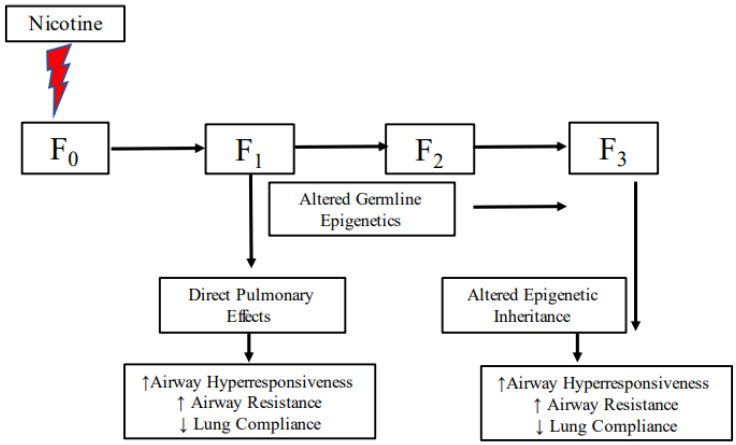
Schematic representation of perinatal nicotine exposure-induced transgenerational transmission of asthma via altered epigenetic inheritance.

## Data Availability

Not applicable.

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
