# Peer review of "Preterm Birth, Developmental Smoke/Nicotine Exposure, and Life-Long Pulmonary Sequelae"

_children, 2023, doi:10.3390/children10040608_

Round 1

Reviewer 1 Report

Manuscript ID: children-2286082

Kurihara, Kuniyoshi and Rehan here provide a very timely and comprehensive review of the effects of smoke and nicotine exposure in preterm infants and the life-long consequences of such exposure. The review is extensive and encompasses the entire spectrum of possible pathogenetic mechanisms involved in the consequences of smoke/nicotine exposure. There are several minor issues that need to be addressed.

1.     Figure 2: The two boxes at the bottom of the figure are not legible.

2.     Page 6, Line 260: Please do not use “etc” – perhaps just add “and” between in vitro fertilization and smoke exposure?

3.     Page 6, Line 283: “..even subsequent generations progeny.” Generations and progeny are redundant words.

4.     Page 7, Line 215: “…thicker alveolar walls, airway smooth muscle deposition, altered…” – this is unclear - smooth muscle consists of cells and is not deposited. Do the authors mean that there is increased SMC hypertrophy, or do they deposit some extracellular matrix?

5.     Page 7, Lines 323 and 324: This sentence (“male’s pulmonary phenotype was more pronounced”) is awkward and needs to be rewritten.

 6.     In multiple places, the authors use jargon such as “preemies” or “ex-preemies” (former preterm infants) and “levels” (concentrations). These should be rewritten.

Author Response

We thank both Reviewers for their positive comments and appreciate their suggestions for further improving our work. As outlined point-by-point below, all suggestions have been incorporated in the revised version.

  1. Figure 2: The two boxes at the bottom of the figure are not Fig 2 legible.

Response: It seems that editorial reformatting of the submitted file resulted in unreadability of the boxed contents. We have confirmed that the contents are legible in both PDF and word files of the revised manuscript.

  1. Page 6, Line 260: Please do not use “etc” – perhaps just add “and” between in vitro fertilization and smoke exposure?

Response: We have made the suggested change.

  1. Page 6, Line 283: “..even subsequent generations progeny.” Generations and progeny are redundant words.

Response: Thank you for pointing this out; the suggested change has been made.

  1. Page 7, Line 215: “…thicker alveolar walls, airway smooth muscle deposition, altered…” – this is unclear - smooth muscle consists of cells and is not deposited. Do the authors mean that there is increased SMC hypertrophy, or do they deposit some extracellular matrix?

Response: Thank you for the constructive feedback. We have clarified the sentence with the referenced publications.

  1. Page 7, Lines 323 and 324: This sentence (“male’s pulmonary phenotype was more pronounced”) is awkward and needs to be rewritten.

Response: Again, thank you for this suggestion – the sentence has been changed as requested.  

  1. In multiple places, the authors use jargon such as “preemies” or “ex-preemies” (former preterm infants) and “levels” (concentrations). These should be rewritten.

Response: The suggested recommendations have been incorporated in the revised manuscript.

Reviewer 2 Report

Preterm Birth, Developmental Smoke/Nicotine Exposure, and Life-long Pulmonary Sequelae by Chie Kurihara et al.

The article shows the paramount impact of smoke lung damage. This article is suitable of publication with a specific interest for preterm birth, as the conclusions go to the wide effects and injuries of the smoke on the public health. 

Author Response

We thank both Reviewers for their positive comments and appreciate their suggestions for further improving our work.